# Exploring the workload of informal caregiving in the context of HIV/NCD multimorbidity in South Africa

**Myrna van Pinxteren**[1]*, **Charlotte Slome**[1], **Frances S. Mair**[2], **Carl R. May**[3,4], **Naomi S. Levitt**[1]

**1** Chronic Disease Initiative for Africa, Department of Medicine, University of Cape Town, Cape Town, South Africa, **2** School of Health and Well-Being, University of Glasgow, Scotland, **3** Department of Health Services Research and Policy, London School of Hygiene and Tropical Medicine, United Kingdom, **4** NIHR North Thames Applied Research Collaboration, United Kingdom

* Myrna.vanpinxteren@uct.ac.za

## Abstract

The importance of informal caregiving for chronic illness has been well established in African contexts but is underexplored in the context of HIV/NCD multimorbidity, particularly in South Africa. Building on treatment burden theories that investigate workload in the context of chronic illness, this paper explores how informal caregiving networks impact the capacity of people living with multimorbidity (PLWMM) in low-income settings in urban and rural South Africa. Qualitative semi-structured interviews were carried out with thirty people living with multimorbidity and sixteen informal caregivers between February and April 2021 in two settings, Cape Town (urban) and Bulungula (rural). Interviews were transcribed and data analysed both inductively and deductively using framework analysis, hereby, building on the principles of the burden of treatment theory (BoTT) as a theoretical lens. Our findings show that informal carers provided different types of support to people living with multimorbidity, including emotional, logistical, health services and informational support to ease the patient's treatment burden. Additional support networks, access to grants or financial security, and feeling a duty to care eased the perceived carer burden while a lack of social support, financial hardship and insufficient information decreased their capacity to support others. Overall, the availability of informal caregiving increases the self-management capacity of people living with multimorbidity in low-income settings in South Africa. Informal caregiving in the context of multimorbidity is structured through social obligations, kinship connections, cultural expectations, and an individual sense of agency. Carers, even when committed to assist, also experience caregiving opportunity costs, which are rarely addressed in the literature. By proposing interventions that can actively support informal caregivers, we can begin to develop solutions that can optimize the role of informal care networks, with a view to improve health-outcomes of PLWMM in South Africa.

**Data Availability Statement:** After consultation with the Human Research Ethics Council (HREC) at the University of Cape Town and the PI – Professor Levitt, it was confirmed we are not able to share

any raw data. The original EXTRA ethics agreement with UCT's HREC and linked consent forms, which were approved in 2020, indicates that we would keep all qualitative research data confidential and would publish interpreted findings and excerpts only. Interested parties can email to peter. delobelle@uct.ac.za or chantal.stuart@uct.ac.za for data related queries.

**Funding:** This work was supported by the United Kingdom Medical Research Council (#MR/ T03775X/1 to NL), the Fogarty International Center (FIC) and the National Institute of Mental Health (NIMH) (#D43TW011308 to MvP for writing support) and the NIHR North Thames Applied Research Collaborative (to CM). The funders had no role in study design, data collection and analysis, decision to publish, or preparation of the manuscript.

**Competing interests:** The authors have declared that no competing interests exist.

# Introduction

Informal caregiving is essential for people living with long term health problems, including those living with multimorbidity–the presence of two or more chronic conditions [1, 2]. Informal care is non-professional, unpaid care provided by family members, friends, neighbours, or community members [3]. Both the practicalities of and need for informal caregiving in African contexts for communicable and non-communicable diseases have been well documented [4– 7]. Caring for vulnerable dependents is a moral bedrock of African traditions, steeped in the Pan-African philosophy of Ubuntu, which encapsulates togetherness, reciprocity, and communality, often seen as opposing the less personal, professional care systems practiced in Western societies [8–11]. In this context, there is also an obligation for multi-generational caregiving within the household, whereby family members are taking care of aging parents, but in addition, grandparents care for grandchildren, even when they are suffering from illnesses themselves [12].

Although caring for others is considered inherent to Africa's social fabric, the impact of the HIV/AIDS epidemic has shown that informal caregiving can be 'a safety net with holes,' as sufficient care is not always available to those affected by stigma, poverty, or an absence of reliable support networks [5, 13, 14]. Furthermore, migration, urbanization, and widening social inequality have had profound effects on household structure and the capacity to care, particularly with the rising rates of non-communicable diseases (NCDs) and increased life expectancy, coupled with an overburdened health system [15–17]. This augments the demands on informal caregivers to provide long-term care for conditions that often become more complex with age. It is within the context of these socio-economic and epidemiological changes that this paper explores the experiences and capacity of carers who are supporting people living with HIV/NCD multimorbidity in urban and rural South Africa.

With more than 8 million people receiving antiretroviral therapy (ART), South Africa has the highest number of people living with HIV (PLWH) in the world, whilst simultaneously experiencing a rising burden of NCDs, most commonly diabetes, cardio-vascular diseases, hypertension, and mental illness [16, 18, 19]. The intersection of these epidemics are increasingly giving rise to multimorbidity, thus increasing patients' workload, and disproportionately impacting economically marginalised groups at a younger age [18, 20]. The consequences of multimorbidity are reflected in limited patient capacity, reduced functionality and productivity, reduced quality of life and increased health costs, leading to poor health outcomes [2]. More than 80% of South Africans use state-funded, public health services, which are fragmented, overburdened, and not equipped to provide the complex care needed by people with multimorbidity [17]. Within this overstretched health system, people living with multimorbidity are expected to take on more responsibility to self-manage their own conditions. This added workload has a significant impact on them and their support networks [21, 22]. Informal caregivers—considered to be part of the support network—have been reported to assist patients with daily self-care, symptom and treatment management, healthy lifestyle behaviours and psychosocial and logistical support [23–25]. However, the level and intensity of informal care given varies per illness, symptom complexity and presence of complications, and is contingent on the people and resources available [4]. This paper explores the complexity of informal caregiving in the context of multimorbidity in South Africa, using interview data collected from people living with multimorbidity and their informal caregivers.

To better understand the experiences and complexities of informal caregiving, data collection and analysis was informed by a theoretical framework, the Burden of Treatment Theory (BoTT) [23, 24]. This theory actively examines the role that support networks and informal caregivers play in the self-management routine of people with chronic illnesses [23]. Defined

as the 'business of being sick,' the BoTT highlights that many care tasks are delegated to informal caregivers [23]. Depending on the complexity of the illness and capacity of the patient, May et al argue that relational networks may become "active partners, co-producers or even 'co-workers' in the organisation, who play a crucial part in delivering the health care work" [23]. As informal caregiving in the context of multimorbidity is currently underreported, this paper explores various types of support provided by caregivers to people living with HIV/NCD multimorbidity in South Africa. We also aim to understand the impact of the context of poverty, rurality, and gender roles on the ability of caregivers to improve capacity of the patient. Using these insights, we then describe ways to better incorporate support networks in an integrated care system whilst increasing the capacity of both patients and caregivers.

## Materials and methods

This study forms part of a larger inter-disciplinary qualitative research project '**EX**ploring the **TReA**tment burden and capacity for self-care amongst people living with HIV/NCD multimorbidity in South Africa to inform the development of interventions to reduce workload and improve capacity' (EXTRA). The EXTRA study aims to develop an adapted theoretical model of HIV/NCD workload-capacity and integrate this model into existing initiatives for patient-centred care, as a basis for developing individual, peer group and service level interventions benefiting people living with multimorbidity in South Africa [26].

### Study design

The study used an in-depth qualitative design to compare caregivers' experiences in two different settings. By design, this study aimed to comparatively explore the differences in treatment workload and caregivers' experiences in two distinct settings where participants experience various degrees of precariousness, including lack of access to services, poverty, inadequate housing, or social isolation [27]. There is also a difference in prevalence and patterns of multimorbidity between rural and urban context, and variations in self-rated health and work productivity [28, 29]. Therefore, multimorbidity can have significant impact on both treatment burden and caregiving experience, which is further explored in this study. Using qualitative methods has proven useful in multimorbidity studies to gain in-depth insights into participant's experiences [30, 31].

### Setting and recruitment

Fieldwork for the EXTRA study was conducted in two locations in South Africa, the urban township of Gugulethu and rural Bulungula. We deliberately chose two comparative field-sites, as we wanted to explore the perceived treatment burden of PLWMM in both rural and urban settings, as access to care, support networks and caregiving experiences may differ. Patient participants were selected using a purposive sampling method with the following criteria: adults living with HIV and at least one other chronic condition who attended public primary health care services. Patient participants were able to bring their primary caregivers, who were interviewed as carer participants, but those without a caregiver present were not excluded from the EXTRA study. Carer participants referenced in this study were either present with the patient participant or mentioned by the patient participant during the interview.

In Cape Town, we selected participants who were enrolled in a linked cross-sectional study that measures the prevalence, patterns and factors of multimorbidity in adults with HIV. Here, 16 patient participants and 10 informal carers were recruited at a community health clinic (CHC) through a trusted fieldworker who also facilitated the consent process The field-site in Bulungula was chosen due to an ongoing relationship with health workers in the area. Here, 14

patient participants and 6 informal carers were recruited from two villages and were receiving treatment at various primary health clinics in the area.

Gugulethu is located 15 km from Cape Town in the Western Cape province and home to more than 100,000. Residents are predominantly isiXhosa speaking and live in formal housing, shacks (informal housing) or as backyard dwellers. Gugulethu is relatively well-serviced, and residents can easily access schools, clinics, and shops. Bulungula is a remote, rural area situated in the Wild Coast region of the Eastern Cape province. The area, a former 'homeland' designated for isiXhosa speaking Black Africans and governed by traditional chiefs, is one of the poorest in the country and has little infrastructure, basic amenities, and few economic opportunities [32]. Population health in the Eastern Cape is characterised by a high burden of both communicable and non-communicable diseases, with an estimated 20% of the population is living with HIV [33]. The prevalence of hypertension is 49.8% [33–35]. Comparatively, an estimated 18% of the Western Cape population is living with HIV and 51.6% with hypertension [33]. The prevalence of diabetes is highest among women in the Eastern Cape (18%), while in the Western Cape it is highest among men (13%) [33].

### Data collection

Data were collected by two post-doctoral researchers, Myrna van Pinxteren (MvP) and Nonzuzo Mbokazi (NM) between February and April 2021. Interviews used a semi-structured interview guide that contained open-ended questions focusing on health information provision, experiences of health services, and social support (S1 Text). The interview guide was adapted from the Burden of Treatment questionnaire (BTQ), and BoTT, aiming to explore patient and carer experiences [23]. An in-depth description of the methodological considerations is published elsewhere [26]. Recruitment and the consent process was assisted by community health workers (CHWs) from the Bulungula Incubator (BI), a local NGO that runs various educational and health programmes for people in the area. Community health workers deliver healthcare services within the communities they are a part of, serving as the first point of contact with the Primary Health Care system [36, 37]. Fieldnotes were generated after each interview containing contextual notes and reflections on the consent and interview process. Interviews were conducted in isiXhosa and English, two of twelve official languages spoken in South Africa. Researchers and participants complied with the required COVID-19 regulations, including masking and social distancing [38].

### Data management and analysis

Before analysis commenced, MvP and NM transcribed and translated all audio-recorded interviews into English. Framework analysis guided the thematic analysis of data, widely used in health and social science research and particularly suitable when researchers code and analyse data in a team [39, 40]. We used framework as an abductive approach, using BoTT and its linked domains to guide our early interpretation of the data. After this, we moved on to thematic analysis to inductively analyse the data. This analysis was an interpretive process led by MvP and Charlotte Slome (CS), using the following steps: familiarisation, development of a coding framework, charting, and further mapping and interpretation [39]. Four carer transcripts were open coded (two urban and two rural) and reviewed by the research team to establish consistency, after which a coding framework was developed [Table 1].

### Ethical considerations

This study followed the guidelines from the Principles of Good Clinical Practice and the Declaration of Helsinki [41]. Ethical approval was obtained from the University of Cape Town (HREC 232/2020) and access to clinics was granted by the Western Cape Department of

**Table 1. Coding framework.**

| Theme | Sub-themes | Definitions |
|---|---|---|
| Carers requirements to control patient's conditions | Learning about conditions and treatments | Have knowledge and understanding of patient's conditions; cope with fears; coping with changing behaviours; monitor patient for signs and symptoms |
| | Assisting the patient with self-care | Assisting with taking medications as prescribed, attending clinic appointments, healthy eating and physical activity |
| How caring is enacted in daily life | Seeking information and understanding to better assist patient | Accessing mass media to learn about patient's conditions; learn from experiences, family/friends, or support groups/NGOs; sharing personal experiences about carer's own co-morbidities |
| | Organise and plan for appointments to assist patient | Assisting with transport; organising funds for clinic visits |
| | Assisting patient to implement regimens | Motivating patient to take medication and create healthy habits; reminding patient about medication schedule; allocating resources for healthier foods and medication |
| | Enlist additional social support | Assisting patient with taking over home duties; mobilising additional relational networks |
| Structural factors impacting carer capacity | Healthcare organisation | Continuity of care; technological support; siloed care |
| | Relationship with health care providers (HCPs) | HCP attitudes toward patients and carers; appreciation for health services |
| | Socio-economic circumstances | Expense of travel, healthy foods, extra medication; high rates of unemployment; dependence on social grants for patient needs |
| | Geographical circumstances | Distance to clinics and hospitals; natural obstacles (river, mountain); rural subsistence farming |
| | Culture | Experiences of stigma when caring; prevalent community values around illness and helping |
| Carer capacity factors that mediate patient burden | Attitude towards illness and circumstances | Extent of fear around patient's conditions; faith and spiritual strength; beliefs around illness |
| | Social support | Availability, type, and extent of additional social support; ability and confidence to seek social support |
| | Ability to overcome barriers to adherence | Resilience; sense of agency; patient's expectations of carer |
| Impact of care burden on quality of life | Emotional well-being | Guilt around the diversion of family resources for patient; need for additional social support |
| | Social | Negotiating social occasions; forming new relationships |
| | Financial | Being forced to move locations (eg. Urban to rural); incurring family debt; loss of income; out-of-pocket expenses |

Health. In the Eastern Cape, we did not recruit respondents through clinics, but received approval from the BI to approach patients identified by the CHWs in their home environments. Potential participants received detailed information sheets outlining the aims of the study, including contact information prior to providing written informed consent in English or isiXhosa. Participant codes and pseudonyms within quotes were used to maintain confidentiality and anonymity.

## Inclusivity in global research

Additional information regarding the ethical, cultural, and scientific considerations specific to inclusivity in global research is included in the Supporting Information (S1 Checklist).

## Findings

In Gugulethu, we interviewed 11 informal caregivers; 6 were patient participants' spouses, 3 were family members (son, daughter, cousin) and 2 were friends. In Bulungula, we interviewed 6 caregivers: 3 daughters, a daughter-in-law, a mother, and a sister wife (a sister through a polygamous marriage). We also conducted a total of 30 semi-structured interviews with patient participants living with HIV and co-morbidities, 16 in Gugulethu, Western Cape and

**Table 2. Participants characteristics Gugulethu, Cape Town.**

| Carer | | | | | | Carer present? | Patient | | | | |
|-------|-----|-----|-----------|----------|-----------|----------------|-------|--------|-----|------------|-----------|
| ID | Sex | Age | Employment | Relation | Conditions | | ID | Sex | Age | Employment | Conditions |
| CU001 | F | 59 | Pensioner | Cousin | Diabetes, hypertension | Yes | PU001 | F | 60 | Unemployed | HIV, diabetes, asthma, hypertension, heart condition |
| CU002 | M | 36 | Unemployed | Son | None | Yes | PU002 | F | 62 | Unemployed | HIV, diabetes |
| CU003 | F | N/A | Unemployed | Daughter | N/A | Yes | PU003 | Female | 53 | Pensioner | HIV, stroke, arthritis, asthma, hypertension, depression |
| CU004 | F | N/A | Employed | Friend | HIV, hypertension | Yes | PU004 | F | 48 | Unemployed | HIV, hypertension, cellulitis |
| CU005 | M | 68 | Pensioner | Partner | Hypertension | Yes | PU005 | F | 63 | Pensioner | HIV, hypertension, TB in hip |
| CU006 | F | N/A | Unemployed | Friend | HIV, hypertension, diabetes | Yes | PU006 | F | 61 | Unemployed | HIV, hypertension, arthirits |
| CU007 | M | N/A | N/A | Husband | HIV | No | PU007 | F | 48 | Unemployed | HIV, hypertension |
| CU008 | F | N/A | Unemployed | Wife | N/A | Yes | PU008 | M | 56 | Unemployed | HIV, hypertension |
| CU009 | F | N/A | Employed | Wife | HIV | Yes | PU009 | M | 46 | Employed | HIV, diabetes, hypertension, depression |
| CU010 | F | N/A | Unemployed | Partner | HIV | Yes | PU010 | M | 57 | Unemployed | HIV, Hypertension |
| CU011 | F | N/A | N/A | Partner | HIV | Yes | PU011 | M | 47 | Self-employed | HIV, hypertension |
| CU012 | N/A | N/A | N/A | Children | N/A | No | PU012 | F | 57 | Self-employed | HIV, hypertension |
| CU013 | M | N/A | Self-employed | Partner | Diabetes, hypertension | Yes | PU013 | F | 65 | Pensioner | HIV, hypertension, diabetes, liver failure |
| | | | | | | | PU014 | M | 72 | Pensioner | HIV, hypertension |
| | | | | | | | PU015 | M | 59 | Unemployed | HIV, hypertension, diabetes |
| CU016 | N/A | N/A | N/A | Sibling | N/A | No | PU016 | M | 46 | Unemployed | HIV, hypertension, stroke |

*F = Female; M = Male; N/A = not available, Grey = not present for interview OR no caregiver identified

14 in Bulungula, Eastern Cape (Tables 2 and 3). The collected findings yielded five different themes; carers providing support to navigate health services, and carers providing emotional, logistical, financial and informational support.

## Providing support to navigate health services

To manage their multiple conditions within South Africa's siloed and unorganised public health system, urban patient participants had to visit various clinics each month. By mobilizing delegated tasks, carers either accompanied patient participants to their appointments or took over their caring responsibilities while the patient was attending the clinic. Due to long waiting times and medication stock-outs, having to wake up in the early hours of the morning to go to the clinic and only being able to leave in the late afternoon was normal.

*"I assist her with coming to the clinic. Sometimes it is three times a month. As she also goes to the physio here as well. So that is also extra. Everything is here [in Gugulethu]."* (CU003F, daughter)

*"I take him and bring him to the clinic, even if it is the whole night, I will wait with him till the morning."* (CU010F, partner)

**Table 3. Participant characteristics Bulungula, Eastern Cape.**

| Carer | | | | | | Carer present? | Patient | | | | |
|---|---|---|---|---|---|---|---|---|---|---|---|
| ID | Sex | Age | Employment | Relation | Conditions | | ID | Sex | Age | Employment | Conditions |
| CR001 | F | N/A | Unemployed | Daughter | N/A | Yes | PR001 | F | 61 | Unemployed | HIV, Diabetes |
| CR002 | F | 18 | Unemployed | Daughter-in-law | N/A | Yes | PR002 | F | 59 | Unemployed | HIV, Epilepsy |
| CR003 | F | N/A | Expanded Public Works Programme (EPWP) | Sister | N/A | No | PR003 | F | 72 | Retired | HIV, hypertension |
| CR004 | F | N/A | N/A | Daughter | N/A | No | PR004 | F | 60 | EPWP | HIV, Heart condition |
| CR005 | F | N/A | Unemployed | Wife | N/A | No | PR005 | M | 41 | Farmer | HIV, Heart disease |
| CR006 | F | 14 | Student | Daughter | TB | Yes | PR006 | F | 42 | EPWP | HIV, Hypertension |
| CR007 | F | Excluded from study, as the patient participant did not meet the inclusion criteria of HIV/NCD multimorbidity | | | | | | | | | |
| CR008 | F | N/A | N/A | Mother | N/A | No | PR008 | F | 40 | Unemployed | HIV, Hypertension |
| CR009 | F | N/A | CHW | Sister-in-law | N/A | Yes | PR009 | F | 42 | Unemployed | HIV, Hypertension |
| CR010 | F | N/A | Student | Granddaughter | N/A | No | PR010 | F | 63 | Unemployed | HIV, Hypertension |
| CR011 | F | N/A | N/A | Mother | N/A | No | PR011 | F | 30 | Unemployed | HIV, Hypertension |
| CR012 | F | N/A | N/A | Sister | N/A | No | PR012 | F | 48 | Unemployed | HIV, Hypertension |
| CR013 | F | N/A | Employed | Sister | N/A | No | PR013 | F | 50 | Unemployed | HIV, Hypertension |
| CR014 | F | 58 | Unemployed | Mother | N/A | Yes | PR014 | M | 34 | Unemployed | HIV, Hypertension, Stomach condition |
| CR015 | F | N/A | Unemployed | Daughter | N/A | Yes | PR015 | F | 63 | Unemployed | HIV, Hypertension, Cancer |

*F = female; M = male; N/A = not available, Grey = not present for interview OR no caregiver identified

The long waiting times experienced occasionally caused distress for carers as they would have to delegate other patient participant's responsibilities, such as childcare.

*"I am not feeling well, because as she is down at the clinic, I keep calling her about what is happening now. And then if things take long, I need to go down to Gugulethu to take care of her child. To watch the child, when she is in hospital [CHC]."* (CU001F, cousin)

For patient participants experiencing mobility problems, carers would assist them collecting medication from the pharmacy, increasing capacity of patient participants. Some patient participants with hypertension were experiencing fatigue, low energy, or headaches, indicating that it was very difficult to get out of bed some days, let alone walk to the clinic and wait for medication. Instead, they would send their carer to the clinic.

*"Most of the time, I am sending my partner to come and get the tablets. Especially in the morning, the pains are worse, and he is coming and fetching my stuff."* (PU005F, 63)

In Bulungula, patient participants received combined care at primary health clinics for both HIV and NCDs, but clinics were far away. In contrast, although clinics were relatively close,

urban participants had to go to different facilities to collect their medication or attend condition-specific appointments. Rural carers would often come along for clinic visits and described the financial and logistical difficulties of this task. They had to travel a minimum of several hours each way to reach the clinic if they were unable to afford the taxi fare. The ferry fare of R20 (1GBP) return to cross the river to reach the clinic was also a significant additional expense, especially for people who did not have access to monthly grants or a stable income.

*"The clinic is far. So, she cannot walk there and come back on the same day. If she is going there, she will have to sleep somewhere and come back the next day. Even when I join her, I have to walk because we cannot afford the taxi."* (CR001F, daughter)

In Bulungula, many carers were responsible for keeping track of patient participants' appointment dates, as most patient participants were illiterate and were unable to read their clinic appointment cards, which resulted in shared decision-making between carer and patient participants. These cards were their only reminder of when the clinic was expecting them, as they did not have consistent access to a mobile phone to receive SMS reminders. Although Gugulethu patient participants and their carers had access to mobile phones, only a few patient participants received technological support from the clinic, including SMS reminders. However, both carer and patient participants felt that receiving more frequent reminders would ease their caring and treatment burdens.

*"We do not get any technological support; it would be useful for him to get a text as a reminder then we can ensure that he does not miss a clinic date."* (CU008F, wife)

## Providing emotional support

Being emotionally involved and motivating patients' adherence and overall well-being was a critical part of informal caregivers' tasks, and patient participants were grateful for their support. However, emotional support and the emotional burden of caregiving were more widely acknowledged in urban dyads. Carers encouraged participants to adhere to their treatment regimens, which may cause side effects, maintain, or adopt healthy behaviours, and uphold their self-worth and dignity, which increased their capacity. For carers who were also HIV+, it was easier to provide emotional support, as they shared similar experiences of stigmatization and prejudice.

*"It is difficult to support him. As he says, he is very stubborn. And I have to fight, literally, because I do understand him, because I also take ARV's. So, I know the road, and must put my foot down, because we are in this together. That's how we support each other."* (CU011F, girlfriend)

When discussing the emotional aspects of informal caregiving, carers expressed a sense of agency and ownership over their caring duties. Although often described as logistically challenging and time-consuming, they also felt proud of their caring abilities. This social obligation to care was expressed mostly by family members.

*"I do not have challenges. It is my duty to do that work. I must take care of her."* (CU002M, son)

*"He is my responsibility because I brought him to this world, I have a duty to him."* (CR014F, mother)

Sharing the workload of caring with other family members also helped carers to cope, which increased carers functional performance and capacity.

*"My husband and all my girls, play a huge part in supporting me and supporting him. We work together as a family to try ensuring that X remains well. Is that not what family is for? Living with conditions is not the end for anyone, but it is that end if they are not supported and motivated and I think that is what all sick people truly want to, know that they are loved and that they matter."* (CR014F, mother)

Other carers would rather take the caring responsibility themselves, even when they would live near other family members.

*"I do not even call other people to help me, like his sister who lives in one of the back rooms. I do it all myself. Sometimes, they will hear later that he was not well. But there is no need to tell her because she probably would not help me anyway. I do it all myself without a problem."* (CU010F, girlfriend)

This carer ascribed her caring abilities to both her personality, but also attributed her caring capacity to her belief system. She relied on God to give her strength.

*"I think I was given the care by God. I never get tired of someone who is sick or get impatient about looking after them. God gave me the ability to care."* (CU010F, girlfriend)

Providing emotional support was not always easy. Some carers expressed exhaustion, especially when they did not have a large support system themselves. This took a toll on their own mental well-being and decreased their functional performance to do the work.

For one carer, balancing the caring responsibility for her mother, providing financial support for the family, and caring for her own 4-year-old son was stressful and taxing. Her mother required extensive care to manage her arthritis, hypertension, HIV, asthma, depression, and impact of a previous stroke. She often felt pressured by her mom to find work whilst continuing her care work, which created tension in their relationship. She had nobody to talk to about these hardships, which were emotionally exhausting, especially as she was also raising her son by herself.

*"It cannot always be me [to take care of my mother], as I have a son and he is 4 years old. And even to her, sometimes he acts out on her. And I need to divide my attention and not become too stressed."* (CU003F, daughter)

However, expressing a degree of resilience, she also believed this was 'God's purpose' and her religious beliefs encouraged her to be strong and reflect on the positives, such as the reciprocation she received from her mom.

"To support my mother is not always easy. But she helps me with my son when she can. And I say thanks to her too. She is my mother." (CU003F, daughter)

Using money she received from her disability grant, the mother would buy clothes and other necessities for her grandson and assist her daughter financially when possible.

### Providing logistical support

Carers also assisted with practical daily household tasks such as cooking, cleaning, or bathing when patient participants were too weak or sick, which increased patient participants' capacity to self-manage their conditions.

*"I cook for her, and I ensure that she takes her pills. I prepare bath water for her. I also wash her clothes for her."* (CR002F, daughter-in-law)

Being supported logistically by their caregivers increased patient participants social capital and functional performance to deal with multimorbidity. One caregiver built a handrail for his partner so she could go to the bathroom independently, as a past TB infection in her hip caused mobility issues and she was wheelchair bound for several years whilst living in a shack. Although she used crutches now, the handrail was still used daily.

*"Even the toilet is difficult for me. So, my partner made a special door for me, a thing, so that when I want to stand up, I must grab there."* (PU005F, 63)

Even with the additions to the house, the partner would still be worried about her safety and asked her not to be too active whilst he was away.

*"And she likes to fall down. [laughs], that is why I tell her every time I leave, don't do anything, just wait for me, as she likes to fall down."* (CU005M, partner)

Managing multiple conditions also meant keeping track of various medication schedules, clinic dates, and health needs. Caregivers would assist patient participants by reminding them to take their medication every day and about their appointments, allowing patient participants to have more control over the services.

*"I pick up the phone to tell her she needs to take the medication, because sometimes she feels lost when she does not feel well.*" (CU001F, cousin)

*"I remind my mother about her clinic dates. I also remind her about her taking her tablets. I even set the alarm for 9 am so that she can take the tablets on time. And I make sure she does not run out of multi-force."* (CU002M, son)

Several carers assisted patient participants with staying on a healthy diet, especially those living with diabetes. One carer enjoyed healthy eating herself and as such, found it easy to assist her partner with adhering to a healthy diet.

*"I really like vegetables, so he has also ended up as someone who eats vegetables a lot... Even if I am in the Transkei with my family... And then tell him that in the fridge we have this and this, continue following our eating pattern. Because we plan our meals. We do not eat the same thing every day. We change things for variety, so he stays healthy."* (CU010F, partner)

Accessing a diversity of food was more challenging for rural participants, as they could not readily access supermarkets. However, all rural participants were able to cultivate seasonal crops such as maize and greens on their land. One carer described how she helped her sister-in-law think of healthy options using the foods she had in her garden, expressing structural resilience and increasing the patient participant's social capital.

*"If she says, 'I do not have food', I will tell her that you have tomatoes, spinach, green pepper, and salt. Get that from your garden and cook and eat."* (CR009F, sister-in-law)

However, rural carers acknowledged that subsistence farming did not produce enough crops to feed their families all year around and would struggle to afford healthy food items or store them, as they did not own refrigerators. High travel costs to the nearest town further hindered their capacity to maintain a healthy diet, which worried carers.

*"She cannot eat nutritious food. Her money does not enable her to be able to eat nutritiously. She eats anything we have."* (CR001F, daughter)

Urban carers described how they mobilised additional support to assist directly with caring or to take care of other family members or responsibilities, so the carer is free to care. One patient participant described how she would ask someone to run her stall and sell food for her when she was not doing well or needed to go to the clinic.

*"But if I am really not well and cannot get up, I ask for help like for someone to sell for me."* (PU012F, 57)

Some urban carers received additional support from a local NGO who would bring groceries or do occupational therapy. Unfortunately, this service was suspended during the pandemic.

*"Before COVID, she had help from social workers. They [Khamya Red Cross, Org] would come and rub her and give her small groceries to ensure older sick people are supported."* (CU002M, son)

## Providing financial support

Mobilizing sufficient resources to assist patient participants weighed heavily on carers, as most carers did not have access to enough exploitable resources such as grant support, which hindered their ability to assist with daily self-management tasks. For example, although care and medication are free at the clinic, not all medication was included, or the quantity would not last a full month. Carers would use household money to purchase out-of-pocket essentials, such as supplements or food.

*"She still needs to buy supplements, multi-force. If she does not get that powder, she will vomit her food. So, every month we get that powder (Multi-force alkaline powder), my brother who is working buys it."* (CU002M, son)

As many carers were in financially precarious circumstances themselves, they would often mobilise additional networks for financial support. Although resources were scarce, carers were creative and resilient, using other family members to chip in.

*"If we don't have enough money, I will take out a loan, but I try my best to ensure we have food, because he cannot take medication without eating."* (CU008F, partner)

*"Like my friend (patient), I am also facing challenges, I am often sick but if I have something I will definitely share it with her. Even with the kids, we have children in their 20s, so when they*

*have money from a job, they will share it so I can eat nutritious food. Her family is like my family.*" (CU006F, friend)

Financial precarity was more prevalent in the rural areas and impacted participants' ability to maintain a healthy diet, buy out of pocket medication or organise transport to the clinic, decreasing patient capacity.

Due to the distances between amenities in the rural areas, the costs associated with traveling to the clinic or hospital were far greater compared to the urban areas. Many carers and patient participants had to borrow money from neighbours to be able to get to their clinic. A strong sense of community and reciprocation allowed this give-and-take borrowing to try and ensure everyone has what they need, which increased social cohesion and improve patients' performance.

*"I borrowed money for us (to travel to the hospital in Mthatha), it is ZAR100 (4 GBP). But that is for transport only, we will not be able to eat."* (CR014F, mother)

*"If my sister cannot give it to me from the Child Support Grant then I must find somewhere I can ask–usually one of our neighbours. Something as simple as going to the clinic is so much effort in the rural setting."* (PR014M, 34)

Hiring a car to go to the clinic or hospital for emergency situations or when the patient is completely immobile is extremely costly for rural participants, ranging between ZAR500-700,00 (20–28 GBP).

*"There is no other way but hiring a car. Like yesterday, I hired a car going to the doctor and hired a car going to the hospital. I simply cannot walk, so it is a lot of expenses for me."* (PR015F, 63)

After losing his job due to a severe gunshot injury, one respondent and his family had to move from an urban city back to rural Bulungula. He was living with HIV and battling a stomach infection. During the interview, he reflected on the guilt and anguish he felt thinking about the amount of debt his family has incurred due to his conditions, who was currently living on two child support grants, which is maximum ZAR1000,00 (40 GBP).

*"I do not even want to know the amount of debt my parents are in; I do not even know how they will ever repay these debts they have incurred because of me and my illness."* (PR014M, 34)

The scarcity of employment opportunities in the rural areas further impacted carers' financial capacity, as they were often unable to leave home to go find work because of their caring responsibilities.

*"I had to make changes in my life to care for my mother, for instance, I cannot go to find work and work."* (CR001F, daughter)

The isolated location of Bulungula also hindered participants accessing other social services, such as Home Affairs or the South African Social Security Agency (SASSA) where people apply for social grants. One carer described how she is unable to receive a child support grant for her three children because her children do not have birth certificates. Although she has made the journey to Home Affairs many times, the problem is still unresolved. The inability to

receive financial support put their family in an extremely precarious position, further limiting her capacity to care for both her children and her mother-in-law.

> "*This is all very frustrating and furthers the struggles we face because money is scarce, we live hand to mouth every day because no one is working a steady job.*" (CR015F, daughter-in-law)

### Providing informational support

Having accurate and sufficient information about one's conditions is vital to successfully manage multimorbidity, as it allows caregivers to make sense of the complexities of chronic illnesses and required support. In the urban areas especially, caregivers recognised the importance of having information about the patient's conditions, which helped them with their caregiving duties.

> "*I had to learn a lot, so I was not getting confused about what medication to give her. And we need to learn about what to do in health emergencies.*" (CU003F, daughter)

Dyads would learn together, through radio, television programmes, or informal conversation, expressing collective sensemaking. Information shared through radio and television programmes often focused on communicable diseases, such as HIV, TB, and COVID-19. Younger caregivers would also utilise Google or social media, but this was less common as it requires data, which many could not readily afford.

> "*I get the information through the radio, social media, and TV. I surf on the internet to learn more about my mom's conditions. That is how I learn to better support people who are living with these conditions.*" (CU002M, son)

Many carers also had one or more chronic conditions themselves, so were able to utilise the knowledge they gained from their own journey to better support the patient. One carer, who was living with HIV, had sufficient knowledge of HIV but relied on her partner for information about hypertension.

> "*With HIV, I got a lot of information from the community, but with HBP, everything I know, I know from him. I learnt it from him. By staying with him, I know what to do.*" (CU011F, partner)

In rural Bulungula, respondents did not have access to a television or electricity and only two of the fourteen had a radio in their home. As such, patient participants and carers relied heavily on others in the community to learn more about their conditions.

> "*Like when we are standing in the line at the clinic, people who have radios will say, I heard this, I heard that, try eating like or eat that. It is not like we can afford to buy the things they tell us about, but we have that sense of togetherness.*" (PR004, female 60)

Information would also be shared by CHWs, who routinely make house visits in Bulungula. One common complaint from both urban and rural carers was the lack of health information provided to them at the clinics, especially for non-communicable conditions. As carers did not always accompany patient participants into the exam room, they were unable to access the information shared during the visit.

*"No, I have not received any information about the conditions. I need additional information and support."* (CR001F, daughter)

*"They (clinics) did not give me any information, but I go to see people from MCSJ [health activism group]. I used to help people, and I learn from others and the information I get through MCSJ and the people you help when you are volunteering."* (CU003F, daughter)

One carer appreciated the counselling she received after her husband was diagnosed with HIV. South Africa has implemented an HIV counselling and testing programme that is mandatory at all public health facilities. HIV or lay counsellors provide information on how to remain healthy and not infect others as well as enrolling the patient into HIV care interventions.

*"Counselling, that was useful, because it really helped him and helped me in preparing on how to live with someone who has HIV positive. Because if you are in a partnership with someone who has HIV, you need the knowledge and counselling on things like how to support them, what they need and not need."* (CU008F, wife)

## Discussion

Our findings describe the various types of support informal caregivers provide to people living with HIV/NCD multimorbidity in South Africa, including emotional, logistical, financial, support to navigate the health services, and informational support. Overall, caregivers felt socially obliged to assist, especially when they were helping family members, but also expressed feeling proud and fulfilled when doing the care work. Patient participants in both the urban and rural field-sites needed support to manage their conditions and were grateful for the assistance carers provided. However, support was structured differently between the two field sites, especially practically, financially, and regarding access to care, whereby accessing health services was more time consuming in the rural area. In Bulungula, carers sought additional support from the whole community, for example, when needing to find refuge on long journeys to the clinic, actively relying on the values of Ubuntu, an essential lifeline during health emergencies or food shortages [8–10, 42]. In urban areas, the care networks were smaller, but caregivers provided more emotional support, which was hardly mentioned by rural carers [8]. These findings point to a difference in social cohesion between urban Gugulethu and rural Bulungula, whereby rural carers and patient participants could rely more easily on their extended networks, whereas some carers and patient participants in Gugulethu felt socially excluded.

Our findings add to the small body of literature exploring informal caregiving in low-middle income contexts and are consistent with previous results [3, 7, 25, 43, 44]. Like other settings, a majority of caregivers were female and were most often the spouse or adult child. All carers experienced opportunity costs, but this was expressed most strongly by younger carers, particularly adult children caring for parents, who needed to navigate looking for employment while caring, which led to feelings of loneliness, especially when they were not able to rely on other family members [3, 25]. Female caregivers experienced the largest workload, as they are culturally expected to fulfil caregiving duties, leading to opportunity costs such as cutting back on paid work or being unable to combine care work with seeking employment [45]. In our study, younger women expressed the emotional strain of their caring responsibilities [25, 45]. Commonly prioritized in other informal caregiving papers is the emotional workload of caregivers, followed by social and financial burdens [42]. In this study, caregivers struggled most with the financial workload and the impact of financial precarity on their caregiving capacity.

This is unique, as socio-economic deprivation and the real-life impact of financial precarity on caring networks is under acknowledged in the literature, especially in studies conducted in high income settings [27, 46, 47]. The wider literature also largely explores the workload of caregiving when caring for patients with short-term, high-demand conditions such as cancer or end-stage organ failure that requires palliative care, whereas our study highlights the under-represented cohort of caregivers caring for people with long-term, low care needs conditions [13, 48, 49].

As South Africa is developing new, integrated primary care systems to improve health service provision for people with chronic illnesses, the role of informal caregiving for those living with multimorbidity is becoming increasingly important, emphasising the ability for patients to self-manage their conditions [50]. Based on the WHO's Care of Chronic Conditions (ICCC) framework, the National Department of Health is rolling out the Integrated Chronic Disease Model (ICDM) in 42 primary care clinics in three of nine provinces [27, 50]. Although still being operationalized, early evaluation studies show that the ICDM fails to address organisational issues, offers insufficient self-management support for patients, and is insensitive towards the increased illness burden faced by people living with multimorbidity [51, 52]. Furthermore, while both ICDM and ICCC models recognise patients' family as stakeholders, no attention is being given to how their role can be better operationalised or supported when providing care for people living with multimorbidity. This is problematic, as initiatives such as the ICDM consider family members to be semi-formalized caregivers who are now responsible for the bulk of the 'chronic homework' stipulated by the health system [53]. Findings in this paper shed light on how the support of informal caregivers is structured and underscores how their involvement increases the capacity of PLWMM.

Using the BoTT as a framework throughout all phases of this study enabled us to explore how informal caregiving is operationalised in our rural and urban South African study sites. We found that the BoTT is a suitable lens through which we can begin to unpack the impact of informal caregiving on the lives of people with multimorbidity in low-income contexts, as participants experienced an increased capacity to deal with their conditions when being assisted with practical tasks, monitoring side-effects and other self-management tasks [23, 24]. Receiving informal care also enhanced participants' structural resilience, the ability to mobilise psychological and social resources to deal with, and even thrive, in the face of adverse social events [23]. The BoTT acknowledges the role of patients' relational networks, but the framework does not distinguish between formal caregiving which is provided by facility or community-based health workers, and informal care, which is non-professional, unpaid care provided by family, friends, neighbours, or community members [3, 13]. This is a lost opportunity, as findings in this paper–one of the few qualitative explorations examining the workload of caregivers in the context of multimorbidity–highlights the opportunity costs of caring, which are not considered within the BoTT and differ from workload experienced by paid health workers. Similar to May et al. (2014), we advocate for 'more interventions that build and strengthen relational networks around patients, and that equip them to work effectively to navigate system controls and opportunity, are therefore likely to improve effective health care utilization' [23]. These interventions are highlighted below and summarized in Table 4.

Firstly, our results show that for caregivers, experiencing a high financial workload was the most stressful, especially when not receiving a stable income, or being unable to seek employment opportunities due to the demands of caring. This, in turn, decreased caregivers' capacity to provide adequate social protection for participants [54, 55]. In Bulungula, caregivers without a regular income expressed that travel costs to the clinic and for emergencies could tip the balance towards household insecurity, resulting in stress and anxiety [27, 55]. To assist carers financially, one solution would be to expand carer-in-aid grant and make the grant more

**Table 4. Overview of possible interventions to unburden carers.**

| Types of support | Attributes impacting on carers' capacity | Proposed intervention | Impact of intervention on increasing patient and caregiver capacity |
|---|---|---|---|
| Emotional | • Emotional burden of caring (loneliness, lack of support system for themselves) | Carer peer-to-peer support groups; access to informal therapy/counselling with peers and CHWs | Helping carers realise they're not alone by connecting them with other carers and providing counselling from CHWs to assist in finding solutions may decrease feelings of loneliness, increase resilience, and may ultimately increase carer capacity. |
| Logistical | • Assisting with daily household tasks<br>• Reminding patient about medication, appointments, required healthy behaviours etc.<br>• Accessing or affordability of healthy foods | Home delivery of medication by CHWs; sending out clinic reminders via SMS | Instituting a medication delivery service would free up carers' time, increasing both patients' and carer capacity. |
| Financial | • Financial precarity/burden limits carer's capacity to provide adequate social protection for patient.<br>• Out of pocket expenses for additional medication, travel costs, additional care, emergencies etc.<br>• Loss of income/work opportunities due to caring responsibilities (opportunity costs)<br>• Inability to buy healthy foods (especially in rural area) | Implementing a form of governmental financial assistance. Either through expanding the dependency grant to include adult care or introducing a Personal Care Budget. | These interventions could ease caregiver workload and increase their ability to mobilise resources, at is allows for more financial freedom to afford healthy foods, extra medications, and emergencies, and carers would feel less pressured to find work. |
| Informational | • Carers not included in health system / patient consultations.<br>• Absence of pamphlets/ informative materials at the clinics for carers | Both health and social policy should focus on strengthening ties between health services, families and communities, a process that can be facilitated by health workers (i.e. family-centred care) | Increased involvement of carers in the health system may increase patient and caregivers sense-making, as they can both learn about medication usage, signs and symptoms to look for, healthy eating options etc. |

flexible to the needs of PLWMM. Currently, the grant is available for those who are already receiving a disability or old age grant, and require full-time care, which many of our participants did not. Additionally, the amount (ZAR510/20GBP) should be increased to match the care dependency grant, which provides ZAR2080 (100GBP) to parents caring for a sick child who needs full-time care. By offering a caregiving grant to informal caregivers of people living with multimorbidity, carers would feel less pressure to find work and could contribute to out-of-pocket medication, travel costs to the clinic, and healthy food options. Worth exploring is also the Personal Care Budget (PCB) [56]. Developed in the Netherlands, the PCB allows carers to get paid for caring tasks that would normally be unpaid labour or tasks done by home nurses, including cleaning, assistance with medication or getting dressed [56]. Acknowledging the differing socio-economic conditions in South Africa, these ideas might inspire new, context-appropriate interventions open to be tailored to assist caregivers of people with multimorbidity.

Secondly, overall, carers felt underinformed, especially about non-communicable diseases, as HIV knowledge was circulated by popular media and in counselling sessions. To increase caregivers' health knowledge, Akpan-Idiok et al. (2020) recommend organising information sessions, or community Imbizo's [gatherings] where knowledge is shared on the intricacies of caregiving for people living with multimorbidity. This intervention showed fruitful among caregivers of cancer patients in Sub-Saharan Africa, who expressed a desire to talk to and learn from others to better understand unique challenges when providing informal cancer care [57].

Thirdly, caregivers in urban areas felt excluded from health services, even if they actively took over self-management tasks for patients, including picking up medication at the clinic. To mitigate this, both health and social policy should focus on strengthening ties between health services, families and communities, a process that can be facilitated by health workers

[3]. Examples of best practices can be found in paediatric medicine, where family-centred care is considered standard [58]. Additionally, caregivers can be allies to health workers when there is a long-term relationship between health workers, patients and informal carers. To foster this, low staff turnover is key and health workers should ask caregivers to join consultations. During clinic visits, caregivers should be encouraged to ask questions and receive emotional support from health workers. To ease the logistic burden of accessing health care, home delivery of medication by CHWs in Cape Town, as initiated during the COVID-19 pandemic, could be reinstated, as an early impact evaluation showed the feasibility and affordability of the intervention, which is also aligned with the commitment to provide community-oriented primary care [59]. Using a broader policy approach, we also need to reimagine the ICDM and other integrated care models and develop new people-centred integrated health frameworks that respond to the challenges of multimorbidity in South Africa and other LMICs [60]. When doing so, we need to encourage participation of informal caregivers and communities in the development of integrated care solutions, as their perspectives are systemically excluded [61].

Lastly, this paper highlighted the emotional workload of informal caregiving, which can lead to stress and anxiety, especially when balancing caring with other responsibilities. In the South African context, where multimorbidity occurs in a younger population, caregiving can be a long-term commitment, which necessitates ongoing emotional support, for instance by engaging with peer-to-peer support groups [62]. This creates an opportunity to establish relationships with other informal caregivers who are navigating the complexities of multimorbidity caregiving. In the Netherlands and Ireland, informal caregivers of dementia patients were offered the opportunity to attend Alzheimer's Cafés. This gathering, whereby attendants received health information and were offered a space to socialize, significantly benefited caregivers in dealing with the daily care for people with dementia, in terms of total well-being, vitality, and emotional workload [56, 57]. Another, African-focused solution might ease the emotional toll on caregivers is the development of Friendship Benches. This mental health intervention, whereby CHWs deliver talk therapy to people with mild to moderate levels of common mental health problems, started in urban Zimbabwe in 2006 to address the shortages of mental health professionals in the country. Evaluation studies showed Friendship Benches have a long-term positive impact on patients' wellbeing, was feasible to sustain overtime and felt rewarding for the CHWs providing care [58, 59]. Giving informal carers access to Friendship Benches to receive additional support from CHWs might be a low-cost solution that can improve carers' ability to manage the emotional burden of informal caregiving.

This study has several limitations. We intended to recruit an equal sample of caregivers in both urban and rural areas, but were only able to recruit six caregivers in Bulungula, as many participants lived alone due to the high rate of temporary and permanent migration, predominantly for work [63, 64]. Compared to the urban interviews with carers, the rural interviews were shorter, as the female participants were hesitant to share details about their caregiving routine, out of respect for elder relatives, even when interviewed alone. Additionally, we only recruited four male caregivers in the study, due to ongoing migration from rural to urban areas and absence of males taking on caregiving roles. The original study focused on the patient participants and as such, full demographic data was not collected for all the caregivers. A key strength is the comparative nature of the study, allowing us to explore the variation within a common experience, as patient participants and carers' lived experiences differed based on their physical access to care, links to the community, and caring burden. However, there were many similarities between caregiver's experiences, regardless of where they lived. Another strength is the prolonged engagement with the study sites, as we travelled regularly to Gugulethu CHC and stayed in Bulungula for two weeks and were able to collect rich insights on people's perceived treatment burden and capacity.

## Conclusion

Findings in this paper compliment the growing body of literature that describes the workload of informal caregiving in South Africa, focusing particularly on HIV/NCD multimorbidity—an underexplored topic. Our work demonstrates that informal caregiving for people living with multimorbidity is time-consuming and hard, especially when living in precarious circumstances and both the burden and capacity of informal caregiving depends on the number of conditions, length of illness, severity of complications and availability of resources. Our paper also emphasizes that the BoTT has viable theoretical underpinnings to investigate caregiving workload in low-income settings, especially when recognizing that both formal and informal caregivers have different opportunity costs and social obligations. We suggest possible interventions that might be adaptable to 'unburden' multimorbidity caregivers in South Africa. By doing so, we showcase the experiences of informal caregivers in the context of multimorbidity and propose what is needed–materially, socially, and practically–to improve the conditions in which care is provided, which is becoming more complex and long-term. In this way, we can begin to develop a comprehensive and adaptable roadmap that can be used to better integrate multimorbidity health services by optimizing role of informal care networks. Whilst acknowledging the differences between settings, this roadmap can be used as a blueprint for envisioning sustainable solutions in multimorbidity care in countries in SSA.

## Supporting information

**S1 Checklist. Inclusivity in global research.**
(DOCX)

**S1 Text. Interview schedule.**
(DOCX)

## Acknowledgments

We want to thank Mr Majola, Dr Boyles and the CHWs from the Bulungula Incubator for their assistance with recruitment. We also want to thank all the participants that were recruited for their time and willingness to share their experiences with us.

## Author Contributions

**Conceptualization:** Myrna van Pinxteren, Charlotte Slome.

**Formal analysis:** Myrna van Pinxteren, Charlotte Slome.

**Funding acquisition:** Frances S. Mair, Carl R. May, Naomi S. Levitt.

**Supervision:** Frances S. Mair, Carl R. May, Naomi S. Levitt.

**Writing – original draft:** Myrna van Pinxteren, Charlotte Slome.

**Writing – review & editing:** Myrna van Pinxteren, Charlotte Slome, Frances S. Mair, Carl R. May, Naomi S. Levitt.

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
