## [Decision Letter · Decision Letter 0]

24 Jan 2024

PGPH-D-23-02163

Exploring the workload of informal caregiving in the context of HIV/NCD multimorbidity in South Africa

Dear Dr. van Pinxteren,

Thank you for submitting your manuscript to PLOS Global Public Health. After careful consideration, we feel that it has merit but does not fully meet PLOS Global Public Health’s publication criteria as it currently stands. Therefore, we invite you to submit a revised version of the manuscript that addresses the points raised during the review process.

We look forward to receiving your revised manuscript.

Kind regards,

Vanessa Carels

Staff Editor

Journal Requirements:

Additional Editor Comments (if provided):

Reviewers' comments:

Reviewer's Responses to Questions

**Comments to the Author**

1. Does this manuscript meet PLOS Global Public Health’s publication criteria? Is the manuscript technically sound, and do the data support the conclusions? The manuscript must describe methodologically and ethically rigorous research with conclusions that are appropriately drawn based on the data presented.

Reviewer #1: Yes

Reviewer #2: Partly

2. Has the statistical analysis been performed appropriately and rigorously?

Reviewer #1: N/A

Reviewer #2: N/A

3. Have the authors made all data underlying the findings in their manuscript fully available (please refer to the Data Availability Statement at the start of the manuscript PDF file)?

Reviewer #1: No

Reviewer #2: No

4. Is the manuscript presented in an intelligible fashion and written in standard English?

Reviewer #1: Yes

Reviewer #2: Yes

5. Review Comments to the Author

Reviewer #1: PLOS Global Public Health

Exploring the workload of informal caregiving in the context of HIV/NCD multimorbidity in South Africa

Spelling mistakes: page 3, 3rd paragraph, line 4; ‘efnancial’—page 12; omission--“….we need encourage participation…”—last sentence of paragraph 6 under ‘discussion”

Study design, recruitment and context: while the study locations are well described and it is perfectly ok to have purposively select them; the authors need to explain what informed selecting those two study locations and even the study provinces out of 9 provinces and literally 1,001 communities that could have been selected. This is critically important to provide more scientificity to the study. Furthermore, it is very important for the authors to explain why they did a rural/urban selection of study sites, and not take it for granted. As the findings show, there appears to be a lot of differences in the data from these two study sites. Was there the need to have rural/urban study sites? Was it planned or not planned? Why was it necessary? This will go a long way to enrich the discussion of the findings.

Data management and analysis: Framework analysis--the authors need to explain, backed by the literature, what this analysis is, and why the need to us it, i.e. the advantages of using it (compared with other useful approaches in the qualitative literature).

Ethical considerations: certain acronyms need to be explained/expanded for the first time: for example, BI, CHWs HCP, EPWP; and CHC.

Tables: The titles/headings for all the tables seem insignificant, based on the font sizes. The font sizes must be made equal to the font in the body of the paper, and each table must have its source and date of generation (e.g., 2022 added to the source).

Obviously, the first table does not fall under the subheading “Ethical considerations”. An appropriate subheading should be created for it or it should be moved and placed before the sub-section on ethical considerations.

Findings

In tables 2 and 3, and in the narrative before these, although you mention studying co-morbidities, and it is one of the basis for having the respondents (if I am not mistaken), several of the respondents have only one morbidity classifications (Hypertension or heart condition or heart disease, etc.). I am not sure how this ties in with the premise of studying co-morbidities.

Page. 10: “As her mother was receiving a disability grant, she would often buy clothes and other necessities for her grandson and assist her daughter financially when possible.” The authors need to clarify if the funds used here is from her mother’s disability grant. It is not readily clear.

Before breaking the findings into the different kinds of support, it is necessary for the authors to give a summary/introduction that the data yielded three different kinds of support, before they detail the three. It reads better this way and prepares the reader to know what to expect.

Discussion

Paragraph 1: there is the need for a more nuanced and a little more detailed rural-urban discussion of the findings than the little that appears at the end of this paragraph.

Paragraph 2: 3rd sentence— “Younger carers, particularly adult children caring for parents, experienced opportunity….” In my mind the opportunity cost would be for every caregiver. This needs to be emphasised and then the authors can further highlight that of the younger caregivers.

“For rural female carers, their role was not always voluntary, but rather a transactional appointment, due to paid lobola (bride price).” This sentence seems a generalisation. Is it that no female carer in the urban area was married to the patient? Is it that all the female carers in the rural area were married to the sick persons? Is it that al female carers in the rural areas who had romantic relationships with the patient had had bride price paid for them? Is it in the rural areas only that married persons paid bride price, and not in the urban areas too, since the respondents from both communities belong to the same tribe. In this respect, does living in rural areas mean paying bride price and in the urban areas mean not paying bride price for marriage? These need to be carefully unravelled.

“…This is unique, as socio-economic deprivation and the real-life impact of financial precarity on caring networks is underreported in the literature, especially in studies conducted in high income settings.” At least two of the literature being referred to here needs to be cited to support this seemingly outstanding statement, to convince the reader.

The first part of the discussion, up to the end of the second paragraph, seems to repeat the findings, without doing due diligence of shedding more light on how the literature quoted in these paragraphs shed light on their findings—e.g. compare, contrast with, support theirs etc. This needs to done, to better inform the reader.

Table heading/titles must uniformly come at the bottom or above the table. Table four has its at the bottom, while the rest do not.

3rd paragraph: First sentence “Using the BoTT as a framework throughout all phases of this study enabled us to explore how informal caregiving is operationalised in rural and urban South African settings.” The highlighted part of the sentence seems a generalisation. The study did not use random samples, and it was restricted to about 2 Provinces, and within these, purposive sampling was used to select the respondents—not even the study sites: hence the findings cannot be generalised to the whole of SA, not even study Provinces, and even the study communities. Conclusions can only be made regarding the respondents in the study, in this case. This correction will have to be done.

4th paragraph, line 2: “…the most stressing for informal…” “…most stressful for…” instead?

5th paragraph: the first two sentences in the paragraph, beginning with “Secondly, during...” seem to belong to the findings section. They were never raised in the findings section, and yet they get discussed. I presume the discussion section is to cover issues raised in the findings, and not to turn the discussion into another section where new findings are provided. The authors should consider finding a more encompassing subtopic for reporting the other findings they leave out at the section on findings and report in the discussions. This is, if they find it important to report these additional findings. Else they can leave them out at the section on discussion as well, for now. They may capture these in another paper, etc.

6th paragraph: “Thirdly, caregivers in urban areas felt excluded from health services…” The same point expressed immediately above seems true about this statement.

“Additionally, health workers should be encouraged to create long-term relationships with informal carers, as they can be allies in caregiving.” It will be very important for the authors to add information on how this could be done—some suggestions of the ‘how to’ will be great. This is to make this recommendation more practical, as the ‘how to’ does not seem to be easy.

Others

The text must make reference to the respective tables.

First sentence under limitations has [REFS]. Not sure if this meant references were to be added. If that is the case, then they need to be provided.

Reviewer #2: Summary

This is qualitative research on the role informal caregivers play in supporting people living with HIV/NCD multimorbidity in South Africa. It is part of a larger study called EXTRA, that explored treatment burden and capacity for self-care amongst people living with HIV/NCD multimorbidity in South Africa. Interviews were conducted with 30 people living with HIV multimorbidity plus with 16 caregivers. The analysis was qualitative framework analysis and utilised two theoretical lenses.

The results are largely descriptive and “describe the various types of support informal caregivers provide to people living with HIV/NCD multimorbidity in South Africa, including emotional, logistical, financial, support to navigate the health services (and) informational support.”

The discussion introduces a number of interesting and relevant points that could be considered results. These include the expanded community support in the rural location, the expectation of women’s work, and the additional financial costs of caring in resource poor settings.

Finally there are 4 interventions suggested and these are mapped to the results

- Financial contribution to carers given the finding of costs of caring

- Information sessions for care-giving – related to findings of information needs

- Inclusion of carers in consultations and medication delivery– related to logistical support

- Emotional support for care-givers – related to providing emotional support

Overall Impression

The paper is interesting and has potential but requires major revisions before being suitable. The organisation of the paper is lacking and there are a number of major issues that need to be addressed.

Overall, I would suggest rethinking the use of the two analytic lenses. Perhaps remove reference to Cumulative complexity model as you don’t seem to have used it in the analysis. Burden of Treatment theory (BoTT) extends the cumulative complexity model anyway so it seems redundant to include both. BoTT talks about capacity components of agency, relationality, control and opportunity which aren’t evident in the framework or the results. Consider revisiting the data analysis using the BoTT components.

There is minimal use of the patient data. I think it would be clearer if you referred to your 16 carers as your participants. Your table of participants should be the CU IDs, and their demographics. Then link to the patient (for whom they care) demographics. If you want to include patient data, it needs to be much clearer, which patient data was used and why and how it was used.

Improvements

Major Issues

Abstract

Review background and methods for consistency with final papers actual objectives.

The result “feeling a duty to care eased the perceived carer burden” is not present in your results and seems a bit incongruous. The first two lines in conclusion are probably results.

Introduction

The fourth paragraph, regarding theory, would be better in methods. Suggest remove reference to Cumulative complexity model.

The fifth paragraph might be better in discussion. The final two lines are the aims of the paper-

“In this paper, we use data derived from an exploratory research project, EXTRA, to define the various types of support provided by informal caregivers. We then describe ways to better incorporate support networks in an integrated care system whilst increasing the capacity of both patients and caregivers.”

It would be better to leave the details about EXTRA to the methods. The aim of the study is to describe the types of support informal caregivers provide to people living with HIV/NCD multimorbidity in South Africa.

I think you also attempt to understand the impact of the context (poverty, rurality, gender roles etc) on the ability of caregivers to improve capacity of the patient, and provide recommendations to support caregivers in this activity.

Such an aim would be consistent with BoTT.

Methods

There is a degree of repetition and confusion in the presentation of the methods. Your own paper on the methodology is structured much better. Use that as a guide in addition to comments.

Suggest a structure similar to below

Setting (or context) includes

- EXTRA study scope and purpose and how your study fits in

- Locations

Then cover Study design

- EXTRA study design including interview design (here instead of in data collection)

- Use of framework of BoTT in design of the study (here instead of in Intro)

- What is the Burden of Treatment questionnaire? Reference required. If you mean TBQ, it should be cited -Tran VT, Harrington M, Montori VM, Barnes C, Wicks P, Ravaud P. Adaptation and validation of the Treatment Burden Questionnaire (TBQ) in English using an internet platform. BMC Med. 2014 Jul 2;12:109. doi: 10.1186/1741-7015-12-109. PMID: 24989988; PMCID: PMC4098922.

- Design of your substudy

The design of your substudy needs to be much clearer. For example, you need to describe who you wanted to recruit, which transcripts would be included and your analytical approach.

Data Collection (and recruitment)

- Include who was eligible here

- How were they recruited

- Who did the interviews and how

Data Management and analysis

- Please include a description of how you followed the Framework analysis steps.

- Table 1 coding framework – seems inconsistent with the results – generally in framework analysis there will be a consistency between framework and the results. Your results are really just theme 2 – how caring is enacted in daily life. See comments below on results.

- The final line seems to belong in study design and needs some further justification.

Ethical consideration

- Recruitment aspects could go in recruitment with only reference to specific ethical considerations here.

Findings

Table 2 and 3 are the participant of EXTRA. For your substudy please show the participants as the carers (plus any patient data you chose to include – with justification in methods).

This will resolve the problem of your quote IDs not being present in the table. Eg CU and CR numbers

The findings are structured according to the 4 activities that carer’s do. There is little to reflect BoTT.

Suggest restructure your analysis around BoTT with a much shorter section on activities done. Eg Table 4, second column, attributes impacting on carer’s capacity could be in results. However I think in BoTT you should be talking about attributes impacting on carer’s ability to support patient capacity.

There is limited comparison of urban and rural – suggest strengthen this if possible (or if not possible remove the wording about comparing in other sections of the paper.

The part on Providing emotional support – seems to be more about emotional impacts of being a carer on the carer. If you re-organise the analysis to focus on attributes impacting on carer’s ability to support patient capacity, the content is appropriate.

Discussion

The discussion first paragraph includes a few findings that are of interest but are unclear in your results- the role of obligation (and see also comment about abstract), patient loneliness if they didn’t have a carer (not sure if this is relevant in your study), Rural support networks (could be much stronger in results).

The second paragraph regarding gender roles is interesting but not very evident in the result. If you redo your table of participants as the carer and include their gender and age it would be useful to support this part of the discussion.

The discussion affirms the suitability of BoTT, perhaps without justification, given its limited use in the results and earlier discussion. BoTT talks about capacity components of agency, relationality, control and opportunity which aren’t evident in the results.

The interventions are good and would be stronger if the results included the attributes impacting on carer’s ability to support patient capacity.

Page 18 – “we only recruited a few male caregivers” – how many should be clear in results

“A key strength is the comparative nature of the study” – need to strengthen comparisons to make this claim.

“reached data saturation’ – how this was determined should be in methods.

Minor Issues

Figures/Tables – Table titles should be at the top of the table

Reference missing end of page 17.

Missing “and” in first sentence of discussion.

Overall a little lengthy, could be more succinct.

6. PLOS authors have the option to publish the peer review history of their article (what does this mean?). If published, this will include your full peer review and any attached files.

**Do you want your identity to be public for this peer review?** For information about this choice, including consent withdrawal, please see our Privacy Policy.

Reviewer #1: No

Reviewer #2: No

---

## [Decision Letter · Decision Letter 1]

17 Jun 2024

PGPH-D-23-02163R1

Exploring the workload of informal caregiving in the context of HIV/NCD multimorbidity in South Africa

Dear Dr. van Pinxteren,

Thank you for submitting your manuscript to PLOS Global Public Health. After careful consideration, we feel that it has merit but does not fully meet PLOS Global Public Health’s publication criteria as it currently stands. Therefore, we invite you to submit a revised version of the manuscript that addresses the points raised during the review process.

Please, carefully follow the comments of the Reviewer.

We look forward to receiving your revised manuscript.

Kind regards,

Adobea Yaa Owusu, MA, PhD, MPH

Guest Editor

Journal Requirements:

2. We have noticed that you have uploaded Supporting Information files, but you have not included a list of legends. Please add a full list of legends for your Supporting Information files after the references list.

Additional Editor Comments (if provided):

Reviewers' comments:

Reviewer's Responses to Questions

**Comments to the Author**

1. If the authors have adequately addressed your comments raised in a previous round of review and you feel that this manuscript is now acceptable for publication, you may indicate that here to bypass the “Comments to the Author” section, enter your conflict of interest statement in the “Confidential to Editor” section, and submit your "Accept" recommendation.

Reviewer #2: (No Response)

2. Does this manuscript meet PLOS Global Public Health’s publication criteria? Is the manuscript technically sound, and do the data support the conclusions? The manuscript must describe methodologically and ethically rigorous research with conclusions that are appropriately drawn based on the data presented.

Reviewer #2: Yes

3. Has the statistical analysis been performed appropriately and rigorously?

Reviewer #2: N/A

4. Have the authors made all data underlying the findings in their manuscript fully available (please refer to the Data Availability Statement at the start of the manuscript PDF file)?

Reviewer #2: No

5. Is the manuscript presented in an intelligible fashion and written in standard English?

Reviewer #2: Yes

6. Review Comments to the Author

Reviewer #2: Thank you for the thorough revision of the manuscript.

There are still some inconsistencies in the methods and findings that need to be addressed.

In Methods

“Setting and recruitment” doesn’t include details of recruitment.

I suggest the section from study design could be second paragraph in setting and recruitment.

“Patient participants were selected using a purposive sampling method with the following criteria: adults living with HIV and at least one other chronic condition who attended public primary health care services. Patient participants were able to bring their primary caregivers, who were interviewed as carer participants, but those without a caregiver present were not excluded from the EXTRA study. Carer participants referenced in this study were either present with the patient participant or mentioned by the patient participant during the interview.”

This could be followed by this section from data collection.

“In Cape Town, 16 patients and 10 ……with the Primary Health Care system (37, 38)”

The following line in study design would be better in data collection (replacing section removed as above).

“The interview guide was adapted from the Burden of Treatment questionnaire (BTQ), and BoTT, aiming to explore patient and carer experiences (23)”

Findings

Table has 15 participants in each site, but your findings say “16 in Gugulethu, Western Cape and 14 in Bulungula, Eastern Cape”

Table 2 patient ID – I think these should be PU not PR numbers?

I like the reformatting of the table to match carers to participants but where there is no Carer interview, it feels a bit misleading to list a CR or CU. I suggest just greying out the CR or CU squares if there is a no carer interview. Also I am unclear what you mean by “dyadic interview”? I thought it was that the carer was interviewed – but you only have 6 yeses in Table 2. What type of interview did the other 4 carers do and can it be indicated on the table?

Eg CU10 – has quotes so was obviously interviewed but there is a no in “dyadic interview” column.

L373 – the quote doesn’t seem to match the findings – loneliness?

7. PLOS authors have the option to publish the peer review history of their article (what does this mean?). If published, this will include your full peer review and any attached files.

**Do you want your identity to be public for this peer review?** For information about this choice, including consent withdrawal, please see our Privacy Policy.

Reviewer #2: No

---

## [Decision Letter · Decision Letter 2]

16 Jul 2024

PGPH-D-23-02163R2

Exploring the workload of informal caregiving in the context of HIV/NCD multimorbidity in South Africa

Dear Dr. van Pinxteren,

Thank you for submitting your manuscript to PLOS Global Public Health. After careful consideration, we feel that it has merit but does not fully meet PLOS Global Public Health’s publication criteria as it currently stands. Therefore, we invite you to submit a revised version of the manuscript that addresses the points raised during the review process.

Please, pay diligent attention to all the Reviewer's comments.

We look forward to receiving your revised manuscript.

Kind regards,

Adobea Yaa Owusu, MA, PhD, MPH

Guest Editor

Journal Requirements:

Additional Editor Comments (if provided):

Reviewers' comments:

Reviewer's Responses to Questions

**Comments to the Author**

1. If the authors have adequately addressed your comments raised in a previous round of review and you feel that this manuscript is now acceptable for publication, you may indicate that here to bypass the “Comments to the Author” section, enter your conflict of interest statement in the “Confidential to Editor” section, and submit your "Accept" recommendation.

Reviewer #2: (No Response)

2. Does this manuscript meet PLOS Global Public Health’s publication criteria? Is the manuscript technically sound, and do the data support the conclusions? The manuscript must describe methodologically and ethically rigorous research with conclusions that are appropriately drawn based on the data presented.

Reviewer #2: Yes

3. Has the statistical analysis been performed appropriately and rigorously?

Reviewer #2: N/A

4. Have the authors made all data underlying the findings in their manuscript fully available (please refer to the Data Availability Statement at the start of the manuscript PDF file)?

Reviewer #2: No

5. Is the manuscript presented in an intelligible fashion and written in standard English?

Reviewer #2: Yes

6. Review Comments to the Author

Reviewer #2: The paper is very much improved and easier to follow.

You haven't fully addressed an inconsistency in the number of patients/carers.

L259, 263 The text indicates 16 patients in Gugulethu and 10 carers but your table 2 has 15 patients and 11 carers. Please rectify.

Minor issues

NA in the tables 2 and 3 is better as N/A.

In the table key it should be N/A = Not available

Grey = not present for interview/no caregiver identified would be better as

not present for interview OR no caregiver identified

References

27 and 51 appear to be the same paper - same DOI. Please rectify.

7. PLOS authors have the option to publish the peer review history of their article (what does this mean?). If published, this will include your full peer review and any attached files.

**Do you want your identity to be public for this peer review?** For information about this choice, including consent withdrawal, please see our Privacy Policy.

Reviewer #2: No

---

## [Decision Letter · Decision Letter 3]

1 Aug 2024

PGPH-D-23-02163R3

Exploring the workload of informal caregiving in the context of HIV/NCD multimorbidity in South Africa

Dear Dr. van Pinxteren,

Thank you for submitting your manuscript to PLOS Global Public Health. After careful consideration, we feel that it has merit but does not fully meet PLOS Global Public Health’s publication criteria as it currently stands. Therefore, we invite you to submit a revised version of the manuscript that addresses the points raised during the review process.

Please, upload a clean copy of the version you revised, as indicated by the Reviewer, and work on all the flagged additional revision needed by the Reviewer.

Please submit your revised manuscript by16th August, 2024. If you will need more time than this to complete your revisions, please reply to this message or contact the journal office at globalpubhealth@plos.org. Please include the following items when submitting your revised manuscript:

We look forward to receiving your revised manuscript.

Kind regards,

Adobea Yaa Owusu, MA, PhD, MPH

Guest Editor

Journal Requirements:

Additional Editor Comments (if provided):

Reviewers' comments:

Reviewer's Responses to Questions

**Comments to the Author**

1. If the authors have adequately addressed your comments raised in a previous round of review and you feel that this manuscript is now acceptable for publication, you may indicate that here to bypass the “Comments to the Author” section, enter your conflict of interest statement in the “Confidential to Editor” section, and submit your "Accept" recommendation.

Reviewer #2: (No Response)

2. Does this manuscript meet PLOS Global Public Health’s publication criteria? Is the manuscript technically sound, and do the data support the conclusions? The manuscript must describe methodologically and ethically rigorous research with conclusions that are appropriately drawn based on the data presented.

Reviewer #2: Yes

3. Has the statistical analysis been performed appropriately and rigorously?

Reviewer #2: N/A

4. Have the authors made all data underlying the findings in their manuscript fully available (please refer to the Data Availability Statement at the start of the manuscript PDF file)?

Reviewer #2: No

5. Is the manuscript presented in an intelligible fashion and written in standard English?

Reviewer #2: Yes

6. Review Comments to the Author

Reviewer #2: Thankyou for adjusting Table 2. The clean copy has not been uploaded but I can see the changes.

Note - The box for CU007 has not been greyed in the same way as CU012.

For N/A - the NA in the table should also be changed to N/A, as well as the in the text below the table. It would be worth confirming that for conditions you mean N/A - or if you mean none - as for CU002 - then write none.

7. PLOS authors have the option to publish the peer review history of their article (what does this mean?). If published, this will include your full peer review and any attached files.

**Do you want your identity to be public for this peer review?** For information about this choice, including consent withdrawal, please see our Privacy Policy.

Reviewer #2: No

---

## [Editor Report · Decision Letter 4]

21 Aug 2024

PGPH-D-23-02163R4

Exploring the workload of informal caregiving in the context of HIV/NCD multimorbidity in South Africa

Dear Dr. van Pinxteren,,

Thank you for submitting your manuscript to PLOS Global Public Health. After careful consideration, we feel that it has merit but does not fully meet PLOS Global Public Health’s publication criteria as it currently stands. Therefore, we invite you to submit a revised version of the manuscript that addresses the points raised during the review process.

**Thank you for your 4^th^ revision. I have carefully reviewed your submitted revision and I still see the following issues with it:**

**1. At pp. 7 and 8, for table 3, information for CROO7 AND PR 007 is skipped. Right from CR/PR006, it moves to CR/PR008, yet there is no explanation for the omission of CR/PR007. Please, insert it in the table or provide an explanation for the omission.**

**2. At page 22, you mention the following Supporting documentation**

980 Appendix 1: Interview schedule EXTRA FINAL DRAFT 30102023

981 Appendix 2: Inclusivity in global research questionnaire

However, I did not see any such supporting documents. Please, provide them.

We look forward to receiving your revised manuscript.

Kind regards,

Adobea Yaa Owusu, MA, PhD, MPH

Guest Editor
---

## [Editor Report · Decision Letter 5]

11 Sep 2024

Exploring the workload of informal caregiving in the context of HIV/NCD multimorbidity in South Africa

PGPH-D-23-02163R5

Dear Dr. van Pinxteren,

We are pleased to inform you that your manuscript 'Exploring the workload of informal caregiving in the context of HIV/NCD multimorbidity in South Africa' has been provisionally accepted for publication in PLOS Global Public Health.

Best regards,

Adobea Yaa Owusu, MA, PhD, MPH

Guest Editor